# Elucidation of 4-Hydroxybenzoic Acid Catabolic Pathways in *Pseudarthrobacter phenanthrenivorans* Sphe3

**DOI:** 10.3390/ijms25020843

**Published:** 2024-01-10

**Authors:** Epameinondas Tsagogiannis, Stamatia Asimakoula, Alexandros P. Drainas, Orfeas Marinakos, Vasiliki I. Boti, Ioanna S. Kosma, Anna-Irini Koukkou

**Affiliations:** 1Laboratory of Biochemistry, Sector of Organic Chemistry and Biochemistry, Department of Chemistry, University of Ioannina, 45110 Ioannina, Greece; e.tsagkogiannis@uoi.gr (E.T.); s.asimakoula@uoi.gr (S.A.); drainas@bio.mx (A.P.D.); orfmar@hotmail.com (O.M.); 2Unit of Environmental, Organic and Biochemical High-Resolution Analysis-Orbitrap-LC-MS, University of Ioannina, 451110 Ioannina, Greece; vboti@uoi.gr; 3Laboratory of Food Chemistry, Sector of Industrial Chemistry and Food Chemistry, Department of Chemistry, University of Ioannina, 45110 Ioannina, Greece; i.kosma@uoi.gr

**Keywords:** biodegradation, *Pseudarthrobacter phenanthrenivorans* Sphe3, 4-hydroxybenzoic acid (4-HBA), protocatechuic acid, catechol, catabolic pathways, metabolomic analysis, plasmid curing

## Abstract

4-hydroxybenzoic acid (4-HBA) is an aromatic compound with high chemical stability, being extensively used in food, pharmaceutical and cosmetic industries and therefore widely distributed in various environments. Bioremediation constitutes the most sustainable approach for the removal of 4-hydroxybenzoate and its derivatives (parabens) from polluted environments. *Pseudarthrobacter phenanthrenivorans* Sphe3, a strain capable of degrading several aromatic compounds, is able to grow on 4-HBA as the sole carbon and energy source. Here, an attempt is made to clarify the catabolic pathways that are involved in the biodegradation of 4-hydroxybenzoate by Sphe3, applying a metabolomic and transcriptomic analysis of cells grown on 4-HBA. It seems that in Sphe3, 4-hydroxybenzoate is hydroxylated to form protocatechuate, which subsequently is either cleaved in *ortho*- and/or *meta*-positions or decarboxylated to form catechol. Protocatechuate and catechol are funneled into the TCA cycle following either the *β*-ketoadipate or protocatechuate *meta*-cleavage branches. Our results also suggest the involvement of the oxidative decarboxylation of the protocatechuate peripheral pathway to form hydroxyquinol. As a conclusion, *P. phenanthrenivorans* Sphe3 seems to be a rather versatile strain considering the 4-hydroxybenzoate biodegradation, as it has the advantage to carry it out effectively following different catabolic pathways concurrently.

## 1. Introduction

4-hydroxybenzoic acid (4-HBA), a phenolic derivative of benzoic acid, is widely used by the food, cosmetic and pharmaceutical industry in the production of preservatives, bactericides, dyes, etc. For example, parabens, the alkyl esters of 4-HBA, have been used in more than 22,000 cosmetic and industrial products, such as dies, pharmaceuticals and animal feed [1,2].

After the introduction of 4-HBA to various environments, it can be highly persistent due to its relatively high chemical stability. This persistence as well as the ever-increasing use of 4-HBA and its derivatives during the past decades have led to their accumulation in the environment, posing potential risks for crops, livestock and human health, making it a priority pollutant and underlying the necessity of its removal [3]. In addition to being a pollutant, 4-HBA is a common intermediate in the catabolism of a plethora of other environmental pollutants, such as phthalates, PAHs and lignin [4,5,6].

To this day, chemical and biological degradations are the most studied means to reduce 4-HBA concentrations in polluted environments, with the latest (biodegradation) being the most sustainable [7,8,9].

The ring cleavages of protocatechuic acid (PCA) at various sites of the aromatic ring are the main catabolic pathways followed for the biodegradation of 4-HBA [7], after the initial hydroxylation to PCA. This reaction is catalyzed by 4-hydroxybenzoate-3-hydroxylase (4HB3H), a class A, flavin-dependent monooxygenase [10]. Subsequently, PCA is cleaved through the *meta*- or the *ortho*-cleavage pathways (catalyzed by 4,5- and 3,4-dioxygenases of PCA, respectively), being eventually funneled into the tricarboxylic acid (TCA) cycle [11]. The *meta*-cleavage pathway has been reported for strains such as *Comamonas* sp. 7D-2 [12], *Delftia* sp. EOB-17 [13], *Comamonas testosteroni* CNB-1 [14] and *Novosphingobium aromativorans* US6-1 [15]. On the other hand, the *ortho*-cleavage pathway has been reported for the strains *Pseudomonas putida* KT2440 [16], *Rhodococcus opacus* [17], *Xanthomonas campestris* [18] and *Cupriavidus necator* JMP134 [19].

Here, an attempt is made to determine the preferred route for 4-HBA catabolism by *Pseudarthrobacter phenanthrenivorans* Sphe3, a strain harboring the genes for 4HB3H as well as for both 4,5- and 3,4-dioxygenases of PCA [11,20]. Although the utilization of 4-HBA as a carbon by *Arthrobacter* sp. has been reported, there is not any study for its degradation pathway [21,22,23]. A better understanding of 4-HBA biodegradation pathways could contribute to the enhancement of the 4-HBA catabolism through condition optimization or even genetic manipulation, leading to cell machineries that could be used for improved bioremediation of polluted environments.

In the present study, we report a metabolomic analysis, supported by transcriptomic data of the wild-type and a mutant strain of Sphe3, having abolished one of the two catabolic plasmids, both grown on 4-HBA as a sole carbon and energy source.

## 2. Results and Discussion

### 2.1. Utilization of 4-HBA as the Sole Carbon and Energy Source

*P. phenanthrenivorans* Sphe3 was originally isolated from a creosote polluted site with the ability to grow on phenanthrene as the sole source of carbon and energy [24]. Sphe3 is a rather catabolically versatile bacterial strain as it is able to grow in the presence of various aromatic compounds, as reported in our previous studies [11,25,26,27]. In the present study, Sphe3 was found to be capable of utilizing 4-HBA as the sole source of carbon and energy. Growth begins immediately as no lag phase is visible (Figure 1). Maximum bacterial growth was observed at 11 h of incubation, while a time-course analysis through HPLC showed the depletion of 4-HBA concentration from the growth medium with a concomitant increase in the cell number, while almost 92% of 4-HBA was already consumed after 12 h of incubation (Figure 1).

In the context of studying the metabolic pathways involved in 4-HBA biodegradation, 4-HBA degradation was studied in both wild-type Sphe3 and the plasmid-cured mutant strain designated as Sphe3c. Sphe3c had lost the large plasmid pASPHE301 (Appendix A), and failed to exhibit growth on phenanthrene as the sole carbon and energy source. However, Sphe3c was able to grow in the presence of 4-HBA by degrading 4-HBA with rates similar to the wild-type strain (Figure 1). The pASPHE301 harbors, among others, genes coding for gentisate dioxygenase and catechol 2,3-dioxygenase (Figure 2). On the other hand, the Sphe3c strain retains the gene for catechol 1,2-dioxygenase, which is located on the bacterial chromosome. By eliminating the plasmid, thus the catechol 2,3-dioxygenase gene, and analyzing the expression levels of catechol dioxygenase genes in both strains, we aimed to clarify if the catechol cleavage pathways are central or peripheral ones in each strain. 

### 2.2. In Silico Analysis of P. phenanthrenivorans Genome: 4-HBA Degradation Pathway

Aerobic biodegradation of 4-HBA varies depending on the bacterial species. Generally, a 4-HBA-3-monooxygenase-hydroxylase (4HB3H, *pob*A) hydroxylates 4-HBA to protocatechuate (PCA), which can then undergo oxygenolytic cleavage by PCA 3,4-dioxygenase (PCD34) to form *β*-carboxy-cis, cis-muconate [4], or by PCA 4,5-dioxygenase (PCD45) to form 4-carboxy-2-hydroxymuconate semialdehyde [6]. Both products are funneled in the tricarboxylic acid cycle. There have also been reports about the alternative pathways of decarboxylation of PCA toward the formation of either hydroxyquinol in *Rhodococcus jostii* RHA1 and *Agrobacterium* sp. [28], or catechol in *Klebsiella pneumoniae* [29], *Klebsiella aerogenes* [30], *Clostridium hydroxybenzoicum* [31] and *Enterobacter cloacae* [32]. Catechol can then undergo the oxygenolytic cleavage of the aromatic ring via *meta*-fission (catalyzed by catechol 2,3- dioxygenase, CDO23) or *ortho*-fission (catalyzed by catechol 1,2-dioxygenase, CDO12) to form 2-hydroxymuconic semialdehyde or *cis,cis*-muconic acid, respectively [25,33]. 

The in silico analysis of the *P. phenanthrenivorans* Sphe3 genome revealed genes that are likely involved in the aerobic catabolism of 4-HBA. Apart from the gene for 4HB3H, interestingly, the Sphe3 genome seems to harbor genes that could be/are involved in both *meta*- and *ortho*-cleavage of PCA as well as *meta*- and *ortho*-cleavage of catechol (Appendix A). More specifically, the genes Asphe3_38690, Asphe3_38850-Asphe3_38860 and Asphe3_35170, located on the chromosome, are considered to encode the 4HB3H and the *α*- and *β*-subunits of PCD34 and CDO12 enzymes, respectively, whereas the genes Asphe3_42380 and Asphe3_40510, located on the plasmids, are considered to encode the PCD45 and CDO23 enzymes, respectively, according to the JGI database (Appendix A). According to a BLASTP search, the protein encoded by Asphe3_38690 shares very high sequence identity (>93%) with 4HB3H from other *Pseudarthrobacter* strains; Asphe3_38850/Asphe3_38860 and Asphe3_42380 share over 90% identity with PCD34 and PCD45 enzymes, respectively, from *Arthrobacter*, *Pseudarthrobacter* and other *Actinobacteria* strains. Asphe3_35170 presents over 91% similarity with other *Arthrobacter* CDO12s, while Asphe3_40510 presents lower identity with other CDO23s. The function of Asphe3_38850/Asphe3_38860 (PCD34), Asphe3_42380 (PCD45) and Asphe3_35170 (CDO12) has been furtherly confirmed earlier by our research group [11,25,26].

There are also numerous genes seemingly coding for transporters belonging to the aromatic acid/H^+^ symport family (MFS superfamily). Among them, gene Asphe3_22000 presents a high amino acid sequence identity (51%, Appendix A) with a 4-HBA transporter from the Gram+ actinobacterium *Subtercola lobariae* [34].

### 2.3. Elucidation of 4-HBA Degradation Pathway in Sphe3

In order to elucidate the 4-HBA degradation pathway in Sphe3, 4-HBA metabolites were extracted from the culture medium in various growth phases and analyzed via LC-MS. Apart from 4-HBA (Appendix A), among the detected intermediate metabolites (Table 1), a compound with *m*/*z* = 153.0193 and a retention time (R.T.) of 1.88 min is identified as PCA based on its MS/MS spectrum as well as its R.T., which are both in agreement with the respective characteristics of the commercially available PCA standard (Appendix A). This result clearly shows that 4-HBA in Sphe3 is channeled through the PCA route and it is in accordance with the expected hydroxylation of 4-HBA by 4HB3H encoded by the Asphe3_38690 gene. Although the most common pathway of 4-HBA degradation is via the formation of PCA, conversion of 4-HBA to gentisate has been reported to occur in some *Bacillus* spp. [35,36] and the archaeal isolate *Haloarcula* sp. strain D1 [37]. In addition, it has been reported that in *Burkholderia xenovorans* LB400, 4-HBA is funneled into the protocatechuate central pathway and potentially into the peripheral gentisate route [38].

A metabolite, which was detected in both the exponential and the stationary growth phases, at a retention time of 0.54 min, with an *m*/*z* of 185.0091 (Table 1 and Appendix A) could coincide with various metabolites of both the *ortho*- and *meta*-cleavage pathways of PCA (Table 1).

As mentioned above, PCA can be *ortho-* or *meta-*cleaved by PCD34 or PCD45, respectively. For the determination of PCD34 and PCD45 activities, enzyme assays with Sphe3 crude extracts from cells grown in 4-HBA were performed, where a higher activity was observed for PCD34 (1.1 U/mL) against that of PCD45 (0.1 U/mL). The expression levels of both *pca*34 and *pca*45 genes confirm that both dioxygenases of PCA are implicated in the 4-HBA degradation pathway, since they are highly and almost equally induced when Sphe3 cells are grown in 4-HBA. In particular, the expression of the *pcd*34 gene seems to be induced by 370 times while that of *pcd*45 is by 450 times, normalized with the respective expression levels when cells are grown in glucose (Figure 3). 

This is the first report of the induction of both *ortho*- and *meta*-cleavage of PCA during the aerobic catabolism of 4-HBA as microorganisms normally accommodate either one or the other. For instance, *Novosphingobium pentaromativorans* US6-1 [15], *Acinetobacter* sp. [39], *C. testosteroni* CNB-1 [14], *Comamonas* sp. 7D-2 [12] and *Delftia* sp. EOB-17 [13] follow the PCA *meta*-cleavage pathway. On the other hand, the *β-*ketoadipate pathway is much more common as it has been reported in numerous bacterial strains such as *R. opacus* [17], *Rhodococcus erythropolis* [40], *P. putida* [16], *Xanthomonas campestris* [41], *Cupriavidus necator* [19], *Glutamicibacter* sp. 0426 [42], *Acinetobacter baylyi* ADP1 [43] and *Variovorax* sp. PAMC26660 [44].

Interestingly, *P. phenanthrenivorans* Sphe3 cells, apart from following the *meta*- and *ortho*-cleavage pathways of PCA, seem to also follow the catechol branch of the *β-*ketoadipate pathway. PCA seems to be also decarboxylated to form catechol, as the LC-MS analysis revealed the presence of a compound at an R.T. of 1.84 min with *m*/*z* = 109.0295, both corresponding to a commercially provided catechol standard (Appendix A). Moreover, a metabolite at an R.T. of 1.34 min with *m*/*z* = 141.0193 was also detected (Appendix A). This *m*/*z* could correspond to several catechol degradation products such as *cis*,*cis*-muconate (CDO12 reaction product) and 2-hydroxymuconate semialdehyde (CDO23 reaction product) (Table 1). Furthermore, both catechol dioxygenase genes were induced when Sphe3 cells were grown in the presence of 4-HBA, a finding that supports the hypothesis that the catechol cleavage pathway is also induced during 4-HBA degradation by Sphe3 cells. However, the expression levels of the *cdo*23 gene showed a striking induction of about 1160 times (Figure 3), compared to the 16-time induction of the *cdo*12 gene, indicating the abovementioned metabolite (with *m*/*z* = 141.0193 and R.T. = 1.34 min) was most likely the CDO23 reaction product, 2-hydroxymuconate semialdehyde.

Another metabolite of the CDO23 cleavage pathway detected in the Sphe3 culture media is 4-hydroxy-2-oxopentanoate (Table 1), a 2-hydroxymuconate semialdehyde hydrolase product, with *m*/*z* = 131.0357 and R.T. = 1.2 min (Appendix A). Since the *cdo*23 gene is located on the pASPHE301 plasmid of Sphe3 (Asphe_40510, Appendix A), a metabolomic analysis of Sphe3c cells that have abolished the pASPHE301 plasmid was carried out. 4-hydroxy-2-oxopentanoate was not detected in the LC-MS analysis of Sphe3c cells grown in the presence of 4-HBA. This result combined with the low induction levels of the *cdo*12 gene in Sphe3c cells (Figure 3) further support the hypothesis that the *meta*-cleavage pathway of catechol prevails compared to the *ortho*-cleavage pathway.

As mentioned above, the presence of catechol among the detected metabolites in Sphe3 cultures in 4-HBA implies the activity of a PCA decarboxylation toward catechol. Even though there are some reports about PCA biotransformation to catechol, this procedure seems to be very rare and to date only a handful of such enzymes have been studied [30,31,32]. A thorough in silico analysis of the Sphe3 genome failed to come up with a PCA decarboxylase sequence. However, Chakraborty et al. recently discovered that in the thermophilus bacterium *Thermus oshimai* JL-2, the PCA decarboxylation to catechol is catalyzed by a 4-carboxymuconolactone decarboxylase [45]. 4-carboxymuconolactone is a metabolite of the PCA 3,4-cleavage pathway and the Sphe3 genome harbors a gene coding for this carboxylase (locus tag Asphe3_38820). 

Unexpectedly, the formation of hydroxyquinol as an intermediate metabolite (*m*/*z* = 125.025, R.T. = 2.06 min) was also observed during the catabolism of 4-HBA (Appendix A) by both Sphe3 and Sphe3c cells, a result consistent with the oxidative decarboxylation of PCA to hydroxyquinol. The formation of hydroxyquinol during the catabolism of PCA has been previously reported for the strains *Rhodococcus jostii* RHA1 and *Agrobacterium* sp. [28], while the hydroxyquinol degradation pathway has been reported in *Burkholderia cepacia* AC1100 [46], *Rhodococcus* sp. strain ON1 [47], *Sphingomonas wittichi* RW1 [48], *Cupriavidus necator* JMP134 [49] and also *Arthrobacter chlorophenolicus* A6, where it seems to be implicated in the biodegradation of 4-nitrophenol [50]. However, a thorough analysis of the genome of Sphe3 did not detect any gene sharing identity with other hydroxyquinol dioxygenases.

## 3. Materials and Methods

### 3.1. Bacterial Strains and Growth Conditions

In the present study, the strains *Pseudarthrobacter phenanthrenivorans* Sphe3 and Sphe3c were used. *Pseudarthrobacter phenanthrenivorans* Sphe3 was isolated from a creosote-polluted area in Epirus, Greece [27], while Sphe3c was constructed in the context of the present study by eliminating the pASPHE301 catabolic plasmid from the wild-type strain (Section 3.2). Both strains were grown in a lysogeny broth (LB) medium or minimal medium M9 (MM M9) as described previously [24], supplemented with 5 mM 4-HBA or 2.2 mM glucose as the sole carbon and energy sources, on a rotary shaker agitated at 180 rpm, at 30 °C.

All chemicals used for the bacterial cultures were purchased from Merck (Darmstadt, Germany).

### 3.2. Curing of pASPHE301 Plasmid

Curing of the pASPHE301 catabolic plasmid was achieved following the method described by Tomoeda et al. [51] with some modifications. Sphe3 cells were grown overnight as described in Section 3.1 in 10 mL of LB supplemented with 0.01% SDS (LB/SDS), which is the minimal inhibitory concentration (MIC) for Sphe3. An aliquot of 100 μL was transferred to 10 mL of LB/SDS and allowed to grow overnight at 30 °C as before. The procedure was repeated seven times in total. At that point, an aliquot of 100 μL was plated in LB agar plates and left to grow at 30 °C for 48 h. 

Fifty colonies were randomly selected and inoculated in 5 mL of LB and were left to grow as described above. The total DNA of each bacterial culture was extracted as described by William et al. [52] and was tested for the presence of the gene of catechol 2,3-dioxygenase (*cdo*23) with specific primer pair (Appendix A). The absence of a product indicates the elimination of the plasmid as this gene is uniquely present in pASPHE301.

Pulsed-field gel electrophoresis (PFGE) had been conducted with total DNA from colonies that did not give a product for *cdo*23. In PFGE, the electric field is not in one direction but interchanges between two different directions, promoting the separation of bigger DNA molecules [53]. 

### 3.3. Preparation of Cell Extracts

Sphe3 and Sphe3c cells were grown in 4-HBA (5 mM) until the mid-exponential growth phase and harvested through centrifugation at 6000× *g* for 15 min at 4 °C. They were subsequently washed with a 50 mM Tris-HCl buffer (pH 8) containing 1 mM DTT and 10 mM PMSF, resuspended in 2 mL of the same buffer and disrupted using zirconium beads (0.1 mm) in a mini bead-beater (Biospec Product, Bartlesville, OK, USA). The homogenate was centrifuged at 12,000× *g* for 40 min at 4 °C and the supernatant was kept in ice to prevent enzyme inactivation until the enzyme assays were performed. Protein concentration of the cell-free extract was determined by applying the Bradford method using the Bio-Rad reagent (Bio-Rad Laboratories, Hercules, CA, USA). Solutions of known concentrations of bovine serum albumin (BSA) (Amresco Inc., Solon, OH, USA) were used for the construction of the standard curve [54].

### 3.4. Enzyme Assays

Protocatechuate 4,5-dioxygenase (PCD45) activity in cell extracts was determined by measuring the increase in absorbance in 410 nm caused by the formation of 2-hydroxy4-carboxymuconate-6-semialdehyde (CHMS) as described elsewhere [11]. One Unit of activity is defined as the amount of enzyme that produces 1 μmol of CHMS in 1 min. The molar extinction coefficient of CHMS used for the calculation of the activity is ε_410_ = 11,200 mM^−1^·cm^−1^.

Protocatechuate 3,4-dioxygenase (PCD34) activity was determined as described by Iwagami et al. [55] with the slight modification of using 50 mM Glycine-NaOH (pH 9.5) as the reaction buffer. The activity was calculated by measuring the decrease in absorbance at 290 nm (PCA substrate consumption), using the molar extinction coefficient for PCA (ε_290_ = 2300 mM^−1^·cm^−1^). One Unit of enzyme activity is defined as the amount of the enzyme that cleaves 1 μmol of PCA in 1 min.

### 3.5. Quantitative Real-Time PCR (RT-qPCR)

Total RNA isolation was performed using the NucleoSpin^®^ RNA Isolation kit by MACHEREY-NAGEL (Düren, Germany) according to the manufacturer’s instructions with slight modifications to the sample homogenization procedure. Specifically, Sphe3 cells were grown in MM M9 supplemented with 5 mM 4-HBA as the sole source of carbon and energy. The cell pellet was obtained by harvesting the Sphe3 culture at the mid-exponential phase of growth and centrifuging at 6000× *g* for 15 min at 4 °C. Then, the cell pellet was resuspended in a 100 μL TE buffer (10 mM Tris-HCl, 1 mM EDTA; pH 8) containing 10 mg/mL of lysozymes using vigorous vortexing. The resulting solution was incubated for 1 h at 37 °C. 

cDNA synthesis was performed using the PrimeScript™ RT Reagent Kit with gDNA Eraser (Perfect Real Time, Takara Bio Inc., Shiga, Japan) according to the manufacturer’s instructions and cDNA was stored at −20 °C. cDNA was further diluted and used as a template at a final concentration of 2.5 ng. The expression levels of the target genes were quantified with RT-qPCR in the CFX Connect Real-Time PCR Detection System (Bio-Rad, USA) using the Kapa SYBR Fast qPCR Kit Master Mix (2×) Universal (Kapa Biosystems, Wilmington, MA, USA), performed as described previously [56]. 

Primers used in the present study are listed in Appendix A. The efficiency (E) of one cycle of RT-qPCR in the exponential phase was found to be 1.87–2.06 (E = 10^(−1/slope)^) [57,58] with correlation factors 0.9901 < R^2^ < 0.9956. The housekeeping gene *gyr*β was used as the reference gene and gene expression levels in glucose were used as a calibrator. The results were analyzed using the relative quantification method [59].

### 3.6. Assessments of Growth and Determination of Residual 4-HBA in Culture Medium

*Pseudarthrobacter phenanthrenivorans* cells, both Sphe3 and Sphe3c, were grown toward the mid of the log phase in the LB medium incubated at 30 °C under agitation. The cells were centrifuged, washed, appropriately diluted and resuspended in 100 mL of the MM M9 medium supplemented with 5 mM 4-HBA as the sole carbon source incubated at 30 °C under agitation in 500 mL flasks. The cultures were inoculated at a starting O.D._600nm_ of 0.15 and bacterial growth was monitored by measuring the optical density at 600 nm at various time points. Cultures inoculated with boiled dead cells were used in parallel as the abiotic negative controls. To monitor 4-HBA removal in Sphe3 and Sphe3c cultures in MM M9 supplemented with 4-HBA as the sole source of carbon and energy, 1 mL of the sample was collected periodically for a time-span of 34 h. Each sample was subsequently centrifuged in 12000× *g* for 1 min and filtered with 20 μm filters and stored in −20 °C until the chromatographic analysis. All determinations were made in triplicate.

### 3.7. HPLC Instrumentation and Chromatographic Analysis Conditions

The chromatographic analysis was carried out using an HPLC system (Agilent, model 1100 series, Agilent Co., Palo Alto, CA, USA). Specific HPLC chromatographic conditions were determined through trial and error using a mixture of the above standard in an effort to optimize the peak resolution, peak height, time of analysis and HPLC gradient elution program using different solvents.

Gradient elution was used at a flow rate of 1 mL/min using (A) an aqueous solution of 0.1% (*w*/*v*) acetic acid, and (B) acetonitrile of an HPLC grade (Merck, Darmstadt, Germany) as the mobile phase. The gradient solvent program was the following: begin with 80% of (A) (0 min) and then decrease to 50% for 30 min (0–30 min), remain at a constant rate for 5 min (30–35 min) and finally increase to 80% at 40 min (36–40 min). 

Separation of the compounds was carried out using a reversed-phase column, Eclipse XDB C18 (150 mm × 4.5 mm × 5 μm, Merck, Darmstadt, Germany), at room temperature. 4-HBA was identified at 280 nm. The samples and standards were prepared on the same day and each sample was analyzed in duplicate (n = 2).

Solutions of known 4-HBA concentration (0.2–10 mM) were used to construct the standard curve. 

### 3.8. UHPLC-LTQ/Orbitrap-HR-MS Analysis

In order to analyze the metabolic profile of Sphe3 and Sphe3c cells grown in the presence of 4-HBA, culture samples of 50 mL were harvested from exponential, late exponential, stationary and late stationary growth phases and centrifuged in 6000× *g* for 30 min to pellet the cells. The supernatants were acidified to pH 2 with HCl 12 M and metabolites were extracted with the solvent extraction method using ethyl acetate. The extracted metabolites were dried under a vacuum and then dissolved in methanol. 

The tentative identification of the metabolites was performed using an Ultra-High-Performance Liquid Chromatography (UHPLC) system coupled to an LTQ/Orbitrap high-resolution mass detector. The system consisted of an automatic sampler (Accela AS autosampler model 2.1.1), an automatic sample flow pump (Accela quaternary gradient U-HPLC-pump model 1.05.0900) and a hybrid LTQ/Orbitrap XL 2.5.5 SP1 mass spectrometer from Thermo Fisher Scientific (Bremen, Germany) equipped with an Ion Max Electrospray Ionization (ESI) probe. The metabolites’ separation was conducted on a reversed-phase Hypersil GOLD analytical column (100 mm × 2.1 mm, 1.9 μm) from Thermo (Bremen, Germany). The mobile phase consisted of phase A (0.1% formic acid and 5 mM FNH_4_ in water) and phase B (acetonitrile). The gradient elution program started at 98% mobile phase A and was maintained for 2 min; the amount of mobile phase B increased to 95% over the next 15 min and stayed constant for an additional 1 min. Then, the mobile phase was restored to 98% A and was left there for 4 min for the column to equilibrate. The flow rate was set at 0.3 mL min^−1^. The injection volume was 10 μL while the tray and oven temperature were set at 15 and 40 °C, respectively.

For the identification of as many metabolites as possible, the MS^2^ conditions were carefully selected and the instrument operated in negative ionization mode. The mass range for the full-scan acquisition mode at 60,000 resolution was set at 100–1200 *m*/*z*. The data-dependent MS/MS mode with parallel acquisition of the top 5 intense ions was performed in the LTQ component and the fragmentation pattern of the isolated adduct ions was obtained by inducing 35% normalized collision energy (NCE). The adopted ESI source conditions were as follows: 45 and 15 arbitrary units (au) of sheath gas and aux gas flow rates, respectively; 320 °C capillary temperature; spray voltage, 3.0 kV; capillary voltage, −30 V; tube lens, −80 V.

The possible metabolites were tentatively identified on the basis of their molecular ion formation ([M-H]^−^) and their characteristic fragments were either explored using the mass spectral library mzCloud™ or were compared to the ones obtained using standard solutions when available, confirming the metabolite identification.

The control of the instrument as the mass spectra process was carried out was with Xcalibur v.2.2 software (Thermo Electron, San Jose, CA, USA).

Water, methanol and acetonitrile solvents (LC–MS purity) were all purchased from Fisher Scientific (Leicestershire, UK). Both formic acid (FA) and ammonium formate (FNH_4_) of 98–100% purity were supplied from Merck (Darmstadt, Germany).

Standard compounds used for the analysis were also purchased from Merck (Darmstadt, Germany).

### 3.9. In Silico Analysis of P. phenanthrenivorans Genome: 4-HBA Degradation Pathway

The genome project is deposited in the Genome Online Database [60] and the complete genome sequence is deposited in GenBank. Sequencing, finishing and annotation were performed by the DOE Joint Genome Institute (JGI) as it is described previously [24]. 

Analyses of catabolic genes potentially associated with 4HBA degradation were carried out with tools within the IMG/MER database. The predicted genes were further compared using BLAST against the KEGG (Kyoto Encyclopedia of Genes and Genomes) database to gain their KOs and pathways.

The BLASTP search of the proteins referred to in the present study was carried out using NCBI standard databases of non-redundant protein sequences (nr), with the protein–protein BLAST algorithm (https://blast.ncbi.nlm.nih.gov/Blast.cgi) (accessed on 27 October 2023). 

## 4. Conclusions

Based on the abovementioned data, Sphe3 can metabolize 4-HBA as a sole carbon source. It is confirmed that 4-HBA degradation in Sphe3 cells exclusively follows the PCA pathway, which is further cleaved via both PCD34 and PCD45. Except for the *ortho*- and *meta*-cleavage of PCA, it is confirmed that in *P. phenanthrenivorans* Sphe3, PCA could be decarboxylated to either hydroxyquinol or catechol. As far as catechol is concerned, it is clear that the cells prefer the *meta*-cleavage of the aromatic ring by CDO23 (Figure 4), whereas further research is required to investigate the hydroxyquinol catabolic pathway. This is the first time that concomitant involvement of the aforementioned pathways is reported. 

## Figures and Tables

**Figure 1 ijms-25-00843-f001:**
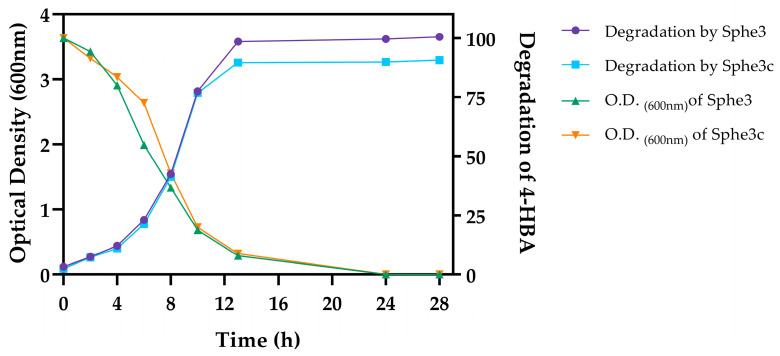
Graphical representation of growth profiles and 4-HBA biodegradation by *P. phenanthrenivorans* Sphe3 and Sphe3c strains. The growth of both strains was measured using optical density at 600 nm while biodegradation rate of 4-HBA in the culture medium by the respective strains was measured using HPLC, expressed as percentage (%) of remaining 4-HBA concentration at various time points.

**Figure 2 ijms-25-00843-f002:**
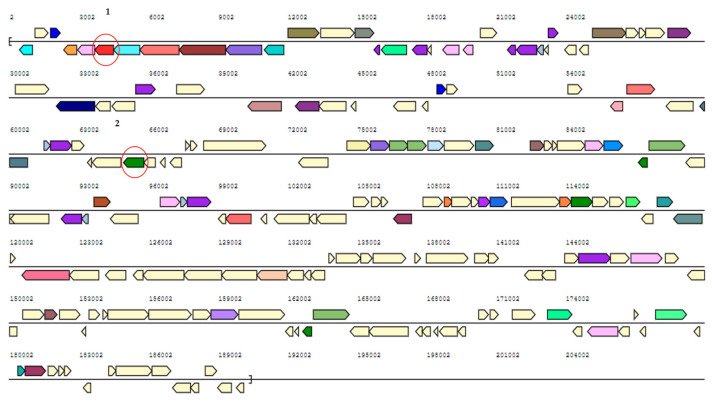
Map of pASPHE301 plasmid from *P. phenanthrenivorans* Sphe3. Marked with red circles are catabolic genes that concern the present study. 1: gentisate 1,2-dioxygenase (*gdo*12, Asphe3_39840), 2: catechol 2,3-dioxygenase (*cdo*23, Asphe3_40510). The plasmid map was extracted from the Joint Genomic Institute (JGI) database (https://img.jgi.doe.gov/cgi-bin/m/main.cgi?section=TaxonDetail&page=taxonDetail&taxon_oid=650377905#genomedetail2) (accessed on 22 July 2023).

**Figure 3 ijms-25-00843-f003:**
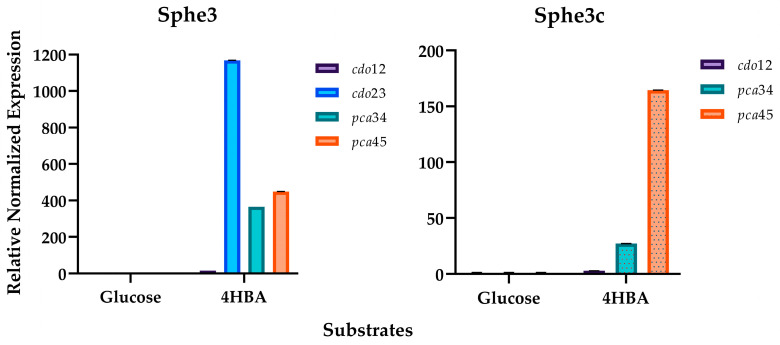
Transcription quantification for the genes involved in the biodegradation of 4-HBA in *P. phenenthrenivorans* Sphe3. The studied genes are coding for catechol 1,2- and 2,3- dioxygenases (*cdo*12 and *cdo*23, respectively) and PCA 3,4- and 4,5-dioxygenases (*pcd*34 and *pcd*45) monitored using RT-qPCR in Sphe3 and Sphe3c cells grown on glucose and 4-HBA, each as the sole carbon and energy source. Values represent the mean relative gene expression normalized to the housekeeping gene *gyr*β ± standard deviations of three individual replicates. Gene expression levels in glucose were used as a calibrator.

**Figure 4 ijms-25-00843-f004:**
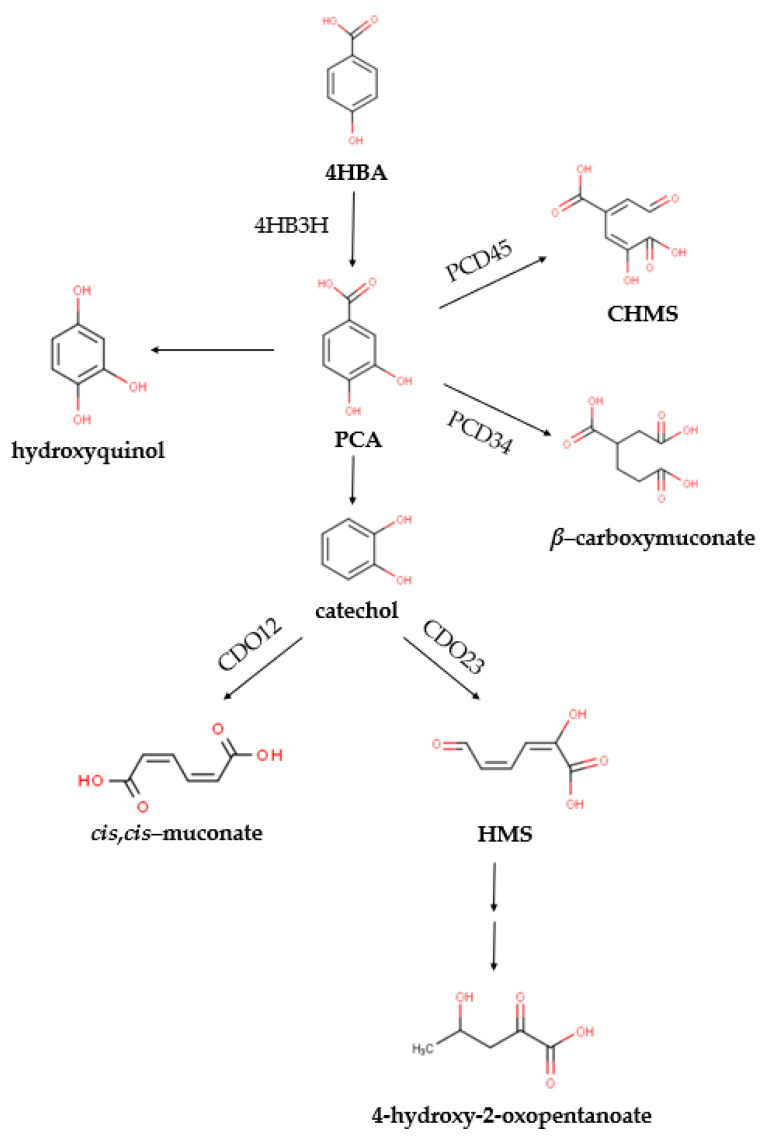
Proposed pathways for the catabolism of 4-HBA in *P. phenanthrenivorans* Sphe3 according to the metabolite analysis using LC-MS. Enzymes expected to catalyze the respective reactions (detected in the Sphe3 genome after the in silico analysis) are mentioned: 4HB3H, 4-hydroxybenzoate-3-hydroxylase; PCD45, PCA 4,5-dioxygenase; PCD34, PCA 3,4- dioxygenase; CDO12, catechol 1,2-dioxygenase; CDO23, catechol 2,3-dioxygenase. All the products are funneled into the central metabolism of the cell (TCA cycle).

**Table 1 ijms-25-00843-t001:** Metabolites detected in various growth phases of Sphe3 or Sphe3c culture in 4-HBA. The catabolic pathways that the detected metabolites could be involved in are designated by the enzyme that catalyzes the respective ring cleavage. CDO12: catechol 1,2-dioxygenase; CDO23: catechol 2,3-dioxygenase; GDO: gentisate 1,2-dioxygenase; HQDO12: hydroquinone 1,2-dioxygenase; PCD34: PCA 3,4-dioxygenase; PCD45: PCA 4,5-dioxygenase; PRCD34: pyrocatechuate 3,4-dioxygenase.

*m*/*z*	(M-H)^−^	R.T. (min)	Compound	Catabolic Pathway	Exponential	Late Exponential	Stationary	Late Stationary
109.0295	C6H5O2	1.84	catechol	-	Sphe3, Sphe3c	Sphe3, Sphe3c	Sphe3c	-
			hydroquinone	-				
125.0250	C6H5O3	2.06	hydroxyquinol	-	Sphe3, Sphe3c	-	-	-
131.0357	C5H7O4	0.91–1.2	4-hydroxy-2-oxopentanoate	CDO23	-	-	Sphe3	Sphe3
137.0244	C7H5O3	3.3	4HB	-	Sphe3, Sphe3c	Sphe3, Sphe3c	Sphe3, Sphe3c	Sphe3, Sphe3c
141.0193	C6H5O4	0.62	cis,cis-4-hydroxymuconate semialdehyde	HQDO12	Sphe3, Sphe3c	Sphe3, Sphe3c	Sphe3, Sphe3c	Sphe3, Sphe3c
			cis,cis-muconate	CDO12				
			(+)-muconolactone	CDO12				
			3-oxoadipate	CDO12				
			2-hydroxymuconate semialdehyde	CDO23				
153.0193	C7H5O4	1.88	PCA	-	Sphe3, Sphe3c	Sphe3, Sphe3c	Sphe3, Sphe3c	Sphe3, Sphe3c
			gentisate	-				
			pyrocatechuate	-				
185.0091	C7H5O6	0.54	4-carboxy-2-hydroxymuconate semialdehyde	PCD45	Sphe3, Sphe3c	Sphe3, Sphe3c	Sphe3, Sphe3c	Sphe3, Sphe3c
			2-hydroxy-2-hydropyrone-4,6-dicarboxylate	PCD45				
			3-carboxy-cis,cis-muconate	PCD34				
			4-carboxymuconolactone	PCD34				
			2-carboxy-2,5-dihydro-5-oxofuran-2-acetate	PCD34				
			maleylpyruvate	GDO				
			3-fumarylpyruvate	GDO				
			3-carboxy-2-hydroxymuconate semialdehyde	PRCD34				

## Data Availability

All data generated or analyzed during this study are included in this published article (and its Appendix A).

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
