# Peer review of "Elucidation of 4-Hydroxybenzoic Acid Catabolic Pathways in Pseudarthrobacter phenanthrenivorans Sphe3"

_ijms, 2024, doi:10.3390/ijms25020843_

Round 1

Reviewer 1 Report

Comments and Suggestions for Authors

This paper describes the biodegradation of 4-hydroxybenzoic acid (4-HBA), by an isolate Pseudarthrobacter phenanthrenivorans Sphe3. Its plasmid cured variant has been experimented in parallel to understand the catabolic pathway that might be functional in this organism for this degradation reaction. There had been several analysis that has been undertaken, including profiling the metabolites and sequencing of the genome (?) / plasmid. There are some major concerns that need to be addressed in the manuscript for clarity, along with few corrections as given.

1. Figure 1, '% remaining 4HBA' is not an appropriate variable. Instead, give degradation (%) data. Also, for the legends, used clear definitions for lines to be differentiated for growth or degradation curves. Though the trend will indicate, yet using isolate name for both the curves may be confusing.    

2. At several places throughout the manuscript, sentences are given in present tense, which should be otherwise in past tense. For methods section specifically.  Overall, the english need to be revised.

3. Line 146-151, its a long statement, and using the term, 'correspond to' is confusing. Rephrase for clarity.

4. Line 109: "Apart from the gene for 4HB3H, interestingly, the Sphe3 genome seems to harbor genes that could be/are involved in both meta- and ortho- cleavage of PCA as well as meta- and ortho-cleavage of catechol.." Now, if these catabolic genes were present in the genome, then why plasmid had been cured to study the pathway. This suggest the possibility of partial diploid systems available (is it so?). This need to be addressed. 

5. line 339 - see centryfiged

6. Fig S2 and tableS2 should be shifted to main manuscript

7. The pathways as given could only be assumed as the analysis is presumptive. Use of C - isotope labeling could have provided the exact intermediates.  

8. The detail methodology for genome sequencing and its analysis is required. The results are given but methods have not been described in the manuscript.

Comments on the Quality of English Language

The language needs to be revised.

Reviewer 2 Report

Comments and Suggestions for Authors

The study aims to reveal the mechanism underlying the metabolisation and decomposition of the 4-hydroxybenzoic acid (4-HBA) type compound by two strains of microorganisms: a strain of Pseudarthrobacter phenanthrenivorans (Sphe3) isolated from a site polluted with creosote, adapted to the use of phenanthrene as a carbon source, and a  mutant strain, derived from Sphe3, obtain by genetic engineering.

Initially, a Sphe3c mutant strain was obtained from Sphe3) after which the two were used in a bioremediation experimental model, in a static environment, on a thermostated shaker, with a minimal culture medium, called M 9 ( or M MM9), which contains 4-HBA as the sole source of carbon. In this experiment, the growth rate of the microorganisms and the metabolisation rate of 4-HBA were followed over time.

The article is interesting, but to be published, the authors must provide additional information and respectively make the necessary corrections as follows:

1) In the Abstract, abbreviations must be avoided

2) The authors must carefully read the entire manuscript and correct the following aspects:

2.1. Reformulation of sentences that begin with an abbreviation (sentences cannot begin with an abbreviation).

2.2. When there are discussions about the figures or tables presented in Supplementary, this must be specified in the text as follows: figure S1 from Supplementary files.

3) The Chapter Materials and Method  must be supplemented with two subchapters as follows

4.8. Lab model used for microbial degradation of 4-HBA

Here authors must provide the volume of the flasks used in the experiment, the number of repetitions, the degree of filling with culture media (25% or...other), and the components (exact composition) of culture media used.

4.9.  In silico Analysis of P. phenanthrenivorans Genome: 4-HBA Degradation Pathway

Here the authors must provide the name of the program used for this study, and eventually, the database (PDB banks type ) used

3) All references from the text must be written unitary, using MDPI rules, with numbers in square brackets (the authors from Rows 44-45 and row 205 must be replaced with numbers in square brackets, and corresponding articles must be added at References).

Round 2

Reviewer 1 Report

Comments and Suggestions for Authors

The authors have addressed the concerns in revised m/s

Author Response

Thank you for your time and your useful comments

Reviewer 2 Report

Comments and Suggestions for Authors

Two sentences beginning with an abbreviation must be rewritten (lines 262 and 333)

Author Response

Thank you for your time and your useful comments. The two sentences have been revised according to your observations.